# Ageing Significantly Alters the Physicochemical Properties and Associated Cytotoxicity Profiles of Ultrafine Particulate Matters towards Macrophages

**DOI:** 10.3390/antiox11040754

**Published:** 2022-04-10

**Authors:** Xu Yan, Yucai Chen, Li Ma, Yongchun Liu, Yu Qi, Sijin Liu

**Affiliations:** 1State Key Laboratory of Environmental Chemistry and Ecotoxicology, Research Center for Eco-Environmental Sciences, Chinese Academy of Sciences, Beijing 100085, China; alvin_yx@163.com (X.Y.); ycchen2022@163.com (Y.C.); sjliu@rcees.ac.cn (S.L.); 2University of Chinese Academy of Sciences, Beijing 100049, China; 3Aerosol and Haze Laboratory, Advanced Innovation Center for Soft Matter Science and Engineering, Beijing University of Chemical Technology, Beijing 100029, China; 2019210873@buct.edu.cn (L.M.); liuyc@buct.edu.cn (Y.L.)

**Keywords:** air pollution, particulate matters, ageing process, macrophage, immune balance

## Abstract

There are still significant concerns about the detrimental effects and health risks of particulate matters (PMs) on the respiratory system. Notably, a largely overlooked knowledge gap is whether the environmental ageing process would change the physicochemical properties of PMs as well as the toxic influences of PMs on macrophages. Here, we applied ambient treatment of model PMs to mimic the real O_3_-induced ageing process and investigated ageing-determined cytotoxicity profile changes of PMs towards macrophages. The consequent distinct bioreactivity and toxicity towards macrophages are largely attributed to the changes of species of surface O-functional groups. Importantly, we unveiled the specific interactions between aged PMs and macrophages due to the variant contents of the surface carboxyl group, resulting in the divergent inflammatory activations and immune balance in the lung. Collectively, this study unearths the significance of ageing in altering particle cytotoxicity, and also provides additional understandings for consecutive investigations on the adverse effects of air pollution on the respiratory system.

## 1. Introduction

Recently, air pollution has raised mounting public concern about health risks, both in academic research and for strategic agendas worldwide [1,2,3]. Specifically, ambient particulate matters (PMs) have been recognized as a major toxic component in air pollution. Upon entering the atmosphere, PMs, especially ultrafine particulate matters (UFPs), will float in the air in the long term due to their small sizes [4,5,6]. It should be noted that the ambient environment contains complicated features, such as SO_2_, O_3_, NO_x_, and so on, and would react with PMs. The long-term process of the aerosol heterogeneous reaction of PMs in the atmospheric environment is often referred to as ageing [7]. Normally, the ageing process alters the physicochemical properties of PMs, e.g., surface functionalities, morphologies, and zeta potentials [8,9,10,11,12]. For instance, ageing by H_2_SO_4_ changes the oxidative potential of PMs [10]. Our previous study also suggested that the ozonation of model PMs gave rise to surface O-functional groups [13]. Although many studies defined the physicochemical changes in PMs after the ambient ageing process, the ageing-determined toxicity changes of PMs are not clearly understood. Therefore, to extend the understanding of the issue, a model PM, i.e., carbon black (CB) particle, and an ageing process in the real atmospheric environment are employed in this study to tease out the mechanisms underlying ageing-associated toxicity changes.

Once inhaled into the respiratory tract, UFPs (i.e., nano-sized PMs or nanoparticles, NPs) could enter and further be cleared by reticuloendothelial cells in the lungs [14,15,16]. Among the various functional cells in the reticuloendothelial systems, macrophages stand out as the first line of defense against foreign particles through playing an important role in particle transport and clearance [17]. As a consequence, macrophages are typical target cells upon UFP invasion in many studies [18,19,20,21]. As documented by previous studies, the mechanisms of cytotoxicity to macrophages caused by UFPs may be ascribed to reactive oxygen species (ROS) generation, the damage of the cell membrane, or inflammatory reactions [22,23,24,25,26,27]. In addition, the oxidative stress and toxicity of UFPs towards cells are largely dependent on the surface properties [28,29]. For instance, Wei et al. showed that the O_3_ aging involved a change in the oxygen-containing functional groups of CB particles, and the content of epoxy groups would induce higher ROS generation and then higher oxidative stress to erythroid cells [13]. Wu et al. observed that the increased oxygen element essentially promoted the biocompatibility of CB particles by elevating the particle dispersion and reducing the aggregation state, which would manifest cytotoxicity inversely related to their oxygen contents [30]. Notably, many recent studies focused on the changes in cytotoxicity upon exposure to UFPs, while the balance in the immune reactions of the macrophages, namely, the polarization states, has been largely overlooked. The polarization states of macrophages can be divided into M1 and M2 states [31]. The M1 macrophages are responsible for constructing a pro-inflammatory micro-environment by secreting cytokines, recruiting more immune cells to clear the invaders. M2 macrophages, on the contrary, will relieve the inflammatory state through inhibiting inflammatory cytokines and thus facilitating tissue repairment [32,33,34]. The balance of the immune state in the lungs is important, and the damage of the balance would cause severe outcomes, such as asthma, pulmonary fibrosis, chronic obstructive pulmonary disease (COPD), and even cancer [29,35,36,37,38,39]. Under this setting, the influence of the intruded UFPs on the macrophages along with the effects towards the immune states of the lungs need to be further investigated.

Therefore, we aimed to investigate the physicochemical alterations of the model UFPs during the ambient ageing process with O_3_, and to interrogate how the ageing process changes the interaction and toxicity profiles of UFPs towards macrophages. In addition, the influences on the immune states of lungs by UFPs with different surface properties were also scrutinized. Together, our combined results unveiled that the alteration in the toxicity effects of UFPs largely resides in the changes in surface physicochemical properties owing to the ageing process.

## 2. Materials and Methods

### 2.1. Ageing Process of CB Particles

Degussa Inc. Corp. (Shanghai, China) provided commercial CB particles (Printex U). The ozonation of CB particles was accomplished using a well-established process, as reported in previous work [10,13]. Briefly, CB particles (30 mg) were firstly dispersed in methanol and then freeze-dried without any heat treatment. Then, the collected powder was spread over a Teflon filter (47 mm) before being exposed to ozone, mixing with ambient air at a total flow of 2.0 L/min. The ozone was generated at a concentration of 996 ppb by an ozone generator (UV-185, Jingxinhe, China), and the exposure time was 2 and 14 h, respectively. The concentration of O_3_ corresponded to 83 ppb, according to the China National Environmental Monitoring Centre (http://www.cnemc.cn/sssj/ accessed on 18 July 2020) in June 2018, in Beijing, China, and the equivalent O_3_ exposure corresponded to the true ageing periods of 1 and 7 days according to Equation (1):

83 (ppb) × 24 (hour) × d (day) = O_3_ conc. (ppb) × t (hour)
(1)


The CB particles were named as CB-0 (pristine CB), CB-1 (ageing for 1 day), and CB-7 (ageing for 7 days), respectively, depending on how long they were exposed to ozone.

### 2.2. Characterization of CB Particles

A scanning electron microscope (SEM, SU-8020, Hitachi, Tokyo, Japan) was used to examine the physical dimensions and morphologies of the CB particles. X-ray photoelectron spectroscopy (XPS, ESCALAB250Xi, Thermo Fisher Scientific, Waltham, MA, USA) was used to determine element components and energy bands, and the results were then evaluated using the program XPSPEAK41. A ZetaSizer Nano ZS (Malvern Nano series, Malvern, United Kingdom) was used to assess the hydrodynamic diameter (*D*_h_) and zeta potential of the CB suspensions (at 10 µg/mL) in deionized (DI) water at 25 °C.

### 2.3. Cell Culture

The murine macrophage cell lines J774A.1 and RAW264.7 were obtained from the Shanghai Cell Bank of Type Culture Collection (Shanghai, China). These cells were cultured at 37 °C in a humidified incubator with 5% CO_2_ and in Dulbecco’s modified Eagle medium (DMEM) supplemented with 100 U/mL of penicillin–streptomycin (Hyclone, Logan, UT, USA) and 10% fetal bovine serum (Gibco, Waltham, MA, USA).

Primary bone marrow-derived macrophages (BMDMs) were obtained from mouse marrows, as described previously [40]. Briefly, BALB/C mice (male, 6 weeks old) were purchased from the Beijing Vital River Laboratory Animal Technology (Beijing, China), and all animal experiment protocols were approved by the Animal Ethics Committee at the Research Center for Eco-Environmental Sciences, Chinese Academy of Sciences (Beijing, China). In short, bone marrow cells were flushed from the femurs and tibias of mice, washed 3 times with PBS, and thereafter cultured in Roswell Park Memorial Institute (RPMI) 1640 medium (Gibco, Rockville, MD, USA) with 100 U/mL of penicillin–streptomycin, 10% fetal bovine serum, and 25 ng/mL of macrophage colony-stimulating factor (M-CSF, PEPROTECH, Chicago, IL, USA) at 37 °C with 5% CO_2_. After incubation for 3 days, the medium was changed every day until day 7. The induced BMDMs were then used for further experiments.

### 2.4. Cytotoxicity Assessment

For cell viability assessment, after seeding in a 96-well (1.0 × 10^4^ J774A.1 cells/well, Corning, NY, USA) plate overnight, J774A.1 cells were treated with CB particles at various concentrations (i.e., 1, 5, 10, 20, 30 μg/mL) for 24 h. Cell viability was then assayed through the cell counting kit 8 (CCK-8) method following the standard manufacturer’s instructions (Solarbio, Beijing, China). Briefly, 10 µL of CCK-8 solution was added to the cell culture medium in each well. The absorbance was then monitored at a wavelength of 490 nm on a plate reader Varioskan Flash (Thermo Fisher Scientific, Waltham, MA, USA) after incubation for 4 h in the dark at 37 °C.

### 2.5. Characterization of the Cells by Confocal Microscopy

After seeding in glass-bottom dishes (3.0 × 10^5^ J774A.1 cells/well, Thermo Fisher Scientific, Waltham, MA, USA) overnight, J774A.1 cells were treated with CB particles at 10 µg/mL for 12 h. Then, the cell membrane was stained with tetramethylrhodamine TRITC Phalloidin (Solarbio, Beijing, China) and the nuclei were stained with 4′,6-diamidino-2-phenylindole (DAPI, Solarbio, Beijing, China), following the instructions from the manufacturer. The fluorescence of the cell membrane (red) and nucleus (blue) were subjected to examination using a TCS SP5 laser scanning confocal microscope (Leica, Wetzlar, Germany). ImageJ software (NIH, Bethesda, MD, USA) was applied to measure the length and width of the cells.

### 2.6. Characterization of Cells by Transmission Electron Microscope (TEM) for Cellular Structure

After seeding in a 6-well plate (3.0 × 10^5^ J774A.1 cells/well) overnight, J774A.1 cells were exposed to CB particles at 10 μg/mL for 12 h and then harvested with phosphate-buffered saline (PBS). Cells were then washed 3 times with PBS and carefully collected after treatment. The collected cells were fixed in 2.5% glutaraldehyde, and thereafter sliced into sections according to the standard protocols. The ultrastructure of the cell was imaged using a Tecnai Spirit TEM at 120 kV (FEI, Waltham, MA, USA).

### 2.7. Intracellular ROS Production Assay of CB Particles

After seeding in a 96-well (1.0 × 10^4^ J774A.1 cells/well) plate overnight, J774A.1 cells were pre-incubated with 10 mM of dichloro-fluorescein diacetate (DCF-DA, Sigma Aldrich, Shanghai, China) probes for 30 min. Thereafter, the cells were washed with PBS at least 3 times prior to exposure to CB particles at 10 and 50 μg/mL for up to 6 h. The DCF-DA fluorescence was then read on a microplate reader (Thermo Fisher Scientific Inc., Waltham, MA, USA) with an excitation wavelength of 488 nm. Untreated cells were used as the negative control and 0.2% H_2_O_2_ was otherwise used as the positive control.

### 2.8. Coomassie Brilliant Blue Staining for Endocytosis Evaluation of CB Particles

After seeding in a 6-well plate (3.0 × 10^5^ J774A.1 cells/well) overnight, J774A.1 cells were exposed to CB particles at 10 μg/mL for 24 h. Thereafter, cells were collected and lysed in PBS with 5% Triton X-100 lysis buffer (Solarbio, Beijing, China) for 30 min. To evaluate the number of CB particles that were endocytosed by macrophages, sodium dodecyl sulfate-polyacrylamide gel electrophoresis (SDS-PAGE) was performed with an equal amount of each lysate (i.e., 60 μg protein content per sample). After electrophoresis, the gel was collected and stained according to the manufacturer’s instructions (Solarbio, Beijing, China). Briefly, the gel was fixed with fixing solution (containing 50% methanol and 10% glacial acetic acid, *v*/*v*) for 1 h, and then stained with staining solution (containing 0.1% Coomassie Brilliant Blue R-250, 50% methanol, and 10% glacial acetic acid, *v*/*v*) for 45 min. After that, the gel was de-stained with the de-staining solution (containing 40% methanol and 10% glacial acetic acid, *v*/*v*) until there was no visible background staining. The protein bands were then recorded, and the ImageJ software (NIH, Bethesda, MD, USA) was applied to differentiate the CB signals in images. CB signals’ intensity was analyzed relative to the CB-0 group.

### 2.9. Determinations of Lactate Dehydrogenase (LDH) Release for Cell Membrane Integrity

After seeding in a 6-well (3.0 × 10^5^ J774A.1 cells/well) plate overnight, J774A.1 cells were exposed to CB particles at 10 μg/mL for 24 h. Cell culture medium was then collected from wells and measured using a kit made by the manufacturer (Nanjing Jiancheng Bioengineering Institute, Nanjing, China).

### 2.10. Real-Time Quantitative Reverse Transcription-Polymerase Chain Reaction (qRT-PCR) Analysis of Gene Expression for Inflammatory Responses

Total RNA was extracted from cells using Trizol reagent, as directed by the manufacturer (Invitrogen, Waltham, MA, USA). The RNA concentration in each sample was measured using a Nanodrop ND2000 instrument (Thermo Fisher Scientific, Waltham, MA, USA). Then, using M-MLV reverse transcriptase (Promega, Madison, WI, USA), 2 µg of RNAs were reversely transcribed into cDNAs. Using a CFX96TM Real-Time Technique, the expression levels of target genes were measured using a standard SYBR Green qRT-PCR system (Bio-Rad Inc, Hercules, CA, USA). The protein cyclophilin (CyC) was utilized to normalize the relative expression of the target genes in this research. The information on the primers is listed in Appendix A.

For pretreatment of the inhibitor against Toll-like receptor 4 (TLR4) signaling, J774A.1 cells were seeded in a 6-well plate (3.0 × 10^5^ J774A.1 cells/well) overnight and then pretreated for 6 h with the inhibitor Resatorvid (Res, 1 μM, MCE, NJ, USA) to suppress TLR4 signaling, followed by CB particles’ exposure at 10 μg/mL for another 24 h. After that, the cells were washed 3 times with PBS before collection for qRT-PCR analysis.

### 2.11. Flow Cytometry Analysis of Macrophage Polarization

After seeding in a 6-well plate (3.0 × 10^5^ J774A.1 cells/well) overnight, J774A.1 cells were exposed to CB particles at 10 μg/mL for 24 h and then harvested with PBS. Post-treatment, the harvested cells were stained with PE-cy7-conjugated anti-mouse CD86 antibody (BioLegend Inc., San Diego, CA United States, 2 μL/10^6^ cells) for 30 min to recognize the polarization states of the macrophages. The Attune NxT Flow Cytometer (Thermo Fisher Scientific, Waltham, MA, USA) was used to perform the flow cytometry analysis. J774A.1 cells without antibody staining were used as the template control.

### 2.12. Statistical Analysis

All data were presented as mean ± standard deviation (SD). The statistical analysis of the experimental data was implemented with an independent *t*-test or one-way ANOVA test. Statistical significance was defined with *p* < 0.05 (*) and *p* < 0.001 (#).

## 3. Results and Discussion

As established, CB particles are commonly used as an ideal model to simulate the real UFPs as CB particles have a similar structure to the elemental carbon of PMs [30,41,42,43]. Here, to simulate the real ageing process in the atmosphere, we conducted the process through exposing CB particles to an ambient environment containing O_3_. The O_3_ concentrations applied in these experiments were determined according to our previous study, in parallel to the real O_3_ levels in June 2018, in Beijing, China [10,13]. The SEM images in Figure 1a–c revealed that the two aged CB particles appeared in uniform particles with a diameter around 30 nm, consistent with CB-0. The SEM images indicated that there was no significant change in both the morphologies and sizes after the ageing process. Nevertheless, the hydrodynamic size exhibited marked variations after ageing, as the *D*_h_ was measured to be 1707.2 nm for CB-0 in DI water solution, and sharply decreased to 515.0 nm for CB-1 and 205.2 nm for CB-7, respectively. Consistently, the changes in *D*_h_ were confirmed by the aggregation states of CB particles in DI water, as the inserted panels showed (Figure 1a–c). Additionally, the surface charge of CB-0 in DI water significantly declined from −8.83 to −20.70 mV for CB-1, and −33.23 mV for CB-7, respectively (Figure 1d), putatively as a consequence of increased surface O-content. These observations, therefore, indicated that the colloidal stability of CB particles increased after ageing with enhanced dispersibility and solubility in water. This is consistent with the increased hydrophilicity of ozonized soot particles compared with the pristine samples, as observed in previous work [44].

Afterwards, the surface functional groups, as characterized by XPS, of the parental and aged CB particles showed considerable changes. Overall, the total surface O-content increased after O_3_ ageing treatment (Figure 1d and Appendix A), in analogy to the previous reports [13]. The O/C ratio increased from 3.54% for pristine CB-0 to 4.42% for CB-1, and 6.81% for CB-7, respectively. Moreover, the various species of O-functional groups on the surface changed a lot after O_3_ treatment. The contents of epoxy/hydroxyl groups (C-O-C/-OH) were elevated to 10.87% in CB-7, compared to the original CB-0 at 8.21% and 8.01% in CB-1. Of note, the carboxyl groups (-COOH) increased from 3.91% for CB-0 to 12.19% for CB-1 and 17.40% for CB-7, respectively. These data thus unraveled the significant alterations in the physicochemical properties of the CB particles, i.e., the oxidative reaction between O_3_ and the carbon backbones [45]. Together, we proposed that the ambient ageing process posed a transformative effect on carbon-based particles in the atmosphere through various mechanisms. It should be noted that the biocompatibility of CB particles is essentially determined by the dispersion properties, and affected by the modification of functional groups of CB particles. Thus, further research is needed to find out if this transformation effect elicited substantial changes to the toxicological effects.

Macrophages are normally recognized as the first-line defenders to clear the invading UFPs, as previously stated [46]. Therefore, to corroborate our hypothesis, macrophages were assayed upon treatment of different CB particles, and J774A.1 cells, as a typical model of macrophages [47], were thereafter chosen for this purpose. As shown in Figure 2, CB-0 and CB-1 particles showed concentration-dependent cytotoxicity to J774A.1 cells, while no significant cytotoxicity was found to J774A.1 cells upon CB-7 treatment. Intriguingly, the greatest cytotoxicity was observed in CB-0-treated J774A.1 cells, with a 42% reduction in cell viability at 30 µg/mL relative to the control group, as evidenced by the CCK-8 assay (Figure 2a, *p* < 0.001). Mild and significant cytotoxicity was demonstrated in CB-1-treated J774A.1 cells, with a 19% reduction in cell viability at 30 µg/mL compared to the untreated control (Figure 2b, *p* < 0.001). Otherwise, negligeable cytotoxicity was found in CB-7-treated J774A.1 cells even at 30 µg/mL, respectively (Figure 2c). In addition, the cytotoxicity profiles of CB particles on another type of murine macrophages, i.e., RAW 264.7 cells, were also evaluated (Appendix A), and the results consistently confirmed the greatest cytotoxicity of CB-0 on macrophages. In support of our data, other researchers have also documented decreased toxicity of carbon-based particles after reduction [30]. For comparison to previous studies, the ROS generation was assayed. CB-0 induced the greatest ROS generation in J774A.1 cells at 50 µg/mL, followed by CB-1 and CB-7 (i.e., CB-0 > CB-1 > CB-7, Figure 3a–c). The results indicated that the CB-0 triggered the greatest oxidative stress in macrophages, consistent with the cytotoxicity profiles. Nevertheless, there was no significant variation in ROS generation upon these different types of CBs, compared with the untreated control under a sublethal concentration (i.e., 10 µg/mL, Figure 3d–f). These findings suggested that a number of complex factors may decide the overall toxicity, as discussed in previous work [13,30]. Intriguingly, despite little ROS generation at the low-exposure concentration, CB particles would still initiate other significant phenotypes in macrophages, such as inflammation, as described above. Considering these findings, the ageing process significantly changed the cytotoxicity profiles of CB particles.

Given that the changes in surface physicochemical properties could influence the toxicity profiles of CB particles, we further conducted experiments to explain the potentially detrimental influence on macrophages through unraveling the interactions between particles and cells. The states of the CB particles and cellular morphologies were observed through confocal microscopy. A sublethal concentration of CB particles (i.e., 10 μg/mL) was used to avoid dramatic cell death. As shown in Figure 4a–d, significant aggregates of CB-0 particles could be recognized in the cytoplasm, indicating the high phagocytosis of CB-0 by J774A.1 cells. Moreover, CB-1 particles could also be found within the cells, but they appeared more scattered instead of large aggregates, corresponding to the change of *D*_h_ after ageing. Otherwise, different interaction patterns between particles and cells were observed for CB-7 treatments. Specifically, limited CB-7 particles were endocytosed by J774A.1 cells, and most of the particles were attached to the plasma membrane (Figure 4c). In the following, the interactions between cells and particles were zoomed in on through TEM observations. As shown in Figure 4e–g, the aggregations of CB-0 and CB-1 were recognized within J774A.1 cells, while CB-7 could only be found around the plasma membrane, in line with the results of confocal microscopy. To substantiate this result, an SDS-PAGE investigation was conducted to semi-quantify the endocytosis of the CB particles with the same number of cells (Appendix A). The trends were in agreement with our observations from both confocal microscopy and TEM (Appendix A). It should be noted that surface O-functional groups not only controlled the direct carbon-based nanomaterial aggregations in the biological medium, but more importantly, the nanomaterial–cell interaction represented a potentially predominant factor in controlling cytotoxicity [48]. When CB particles were highly oxidized, namely CB-7, the colloidal stability would increase (Figure 1d). However, the higher content of the carboxyl group (-COOH) on the surface appeared to transform CB particles into a likely “edge-to-edge” manner, as described previously [48,49]. The special form of aggregation morphology of CB-7 particles would inhibit the endocytosis of macrophages and then diminish the cytotoxicity effects, in agreement with the previous study [49].

Based on the special interaction between different CB particles and macrophages, the LDH releases after various exposures were assessed. The results showed that CB-7 induced the greatest LDH release of J774A.1 (Figure 5a,b), suggesting obvious impairments of the plasma membrane after CB-7 treatment. Remarkably, according to the confocal microscopy, the cell morphologies changed significantly after exposure to the original and aged CB particles, as evidenced by the ratio of length to width (Figure 5c). The macrophages after CB-7 treatment incurred a more obvious differentiation state, and the ratio of length to width of the macrophages was higher than that of the other groups, CB-0 and CB-1 (*p* < 0.05). As documented, upon stimuli, macrophages would tend to be in an activation state, and the increase in length/width ratio is a typical marker [50]. Based on the changes in cell morphologies, we deduced that the CB-7 would induce more severe inflammatory reactions and M1 polarization of macrophages. To this end, the mRNA expression of two typical M1 markers, i.e., interleukin-1beta (IL-1*β*) and interleukin-6 (IL-6), was investigated. As shown in Figure 6a,b, the CB-7 particles enhanced the expression of IL-1*β* during 24 h exposure (with approximately 10.45-fold relative to the untreated cells, *p* < 0.001). In addition, similar changes in IL-6 were demonstrated, supporting the findings on the robust inflammation reactions upon CB-7 treatment. For comparison, the expressions of IL-1*β* and IL-6 induced by CB particles in other model macrophages, i.e., RAW 264.7 cells and BMDMs (namely the murine primary culture cell), were also investigated, and the results were in parallel to those of J774A.1 cells, as the CB-7 induced the most severe inflammatory reactions (Appendix A). In agreement with the above data, the flow cytometry analysis also showed that the CD86 expression, namely the M1 marker, increased in macrophages in response to CB-7 treatment for 24 h, greater than CB-1 (Figure 6c).

Supported by our previous studies, carbon-based nanomaterials can trigger pro-inflammatory responses of macrophages through the activation of TLR4 on the cell membrane [51]. According to the interaction of CB-7 with the cell membrane, we thereafter examined the roles of the TLR4 signaling pathway in dictating M1 polarization by CB-7. Therefore, Res, as a selective inhibitor against TLR4, was used to block TLR4 activity, according to a well-established method in a previous study [52]. As shown in Figure 6d,e, Res markedly repressed the expression of IL-1*β* and IL-6 by approximately 84% and 83%, respectively, in macrophages upon CB-7 exposure (*p* < 0.001), suggesting that TLR4 signaling was closely implicated in the CB-7-induced inflammatory response. Collectively, CB-7 particles were able to induce the M1 polarization of macrophages, and may drive the immune micro-environment of the lung into an inflammatory state.

Of various air pollutants, ambient fine particles, especially UFPs (i.e., PM_0.1_), are thought to be the most concerning in terms of their detrimental health effects on the lung, since they can escape from the mucous barrier and enter the lungs [18,53]. In an effort to find out the influences of the ageing process on the cytotoxicity effects of UFPs, we carried out experiments to mimic the real O_3_ oxidation process and figured out the surface O-functionalities as one physicochemical determinant for changing the toxicity profiles. In line with previous studies, the ageing of CB particles would greatly change the stability and species of surface O-containing groups [13]. Given that no significant changes were observed in morphology and size, thus, here we focused on one important variable affecting cytotoxicity, i.e., the changes in surface functional groups upon O_3_-induced ageing. Based on our results, a trend of gradient cytotoxicity in macrophages, namely CB-0 > CB-1 > CB-7, could be concluded. Although the CB-7 particles exhibited the least cytotoxicity, CB-7 could induce the most serious inflammatory reactions in macrophages. Intriguingly, our results indicated no significant variation in ROS generation upon these different types of CBs under a sublethal concentration. Nonetheless, CB particles still induced other detrimental outcomes in macrophages, e.g., inflammation, especially CB-7. The remarkable biological effects of CB-7 might be due to the significant increase in carboxyl groups on the surface after ageing, increasing colloidal stability [13]. Thereby, the high content of carboxyl groups would also transform the aggregation state of CB-7, leading to less endocytosis of CB-7 particles by macrophages, analogous to the previous report [48]. Otherwise, this transformation of CB-7 particles would increase the association of particles onto the plasma membrane and consequently evoke the M1 polarization of macrophages through the TRL4 pathway. The CB-7 exposure would eventually induce severe inflammation outcomes and compromise the immune balance in the lung. Notably, the alterations of the immune balance in the lung have been proposed as reasons for various diseases, such as COPD, acute lung injury, and pulmonary hypertension, as documented [54,55,56]. Thus, these results unraveled a previously unrecognized role of surface properties in determining cytotoxicity, that is, the alterations in surface O-functionalities after the ageing process can alter the interaction patterns of particles with macrophages, and then change the cytotoxicity and the immune balance in the lung, as illustrated in Figure 7. It should also be noted that our study focused on the in vitro experiments, and the findings did warrant follow-up in vivo studies. Nevertheless, our findings stressed the importance of the ambient ageing process on cytotoxicity and, more importantly, long-term inflammation responses and the balance of the lung immune micro-environment.

## 4. Conclusions

We reported the impact of the ageing process on the physicochemical changes of UFPs, and also the cytotoxicity profiles and the inflammatory reactions in macrophages. Airborne UFPs are subjected to various atmospheric conditions, such as air pollutants (e.g., SO_2_, O_3_, and NO_x_), light, and humidity, giving rise to remarkable alternations in surface O-functionalities. Under this premise, despite numerous toxicity studies on fine air particles, rather limited knowledge has been obtained to understand ageing-determined toxicity profiles. Airborne particles after inhalation can enter the lung deeply and then be endocytosed by macrophages. Our data unraveled that the species of surface O-containing groups essentially dictated the bioreactivity and cyto-compatibility of aged CB particles towards macrophages, and even the inflammatory reactions in macrophages. These findings, therefore, offer a new point of view in proving the risks of particulate pollution and the effects associated with the ageing process.

## Figures and Tables

**Figure 1 antioxidants-11-00754-f001:**
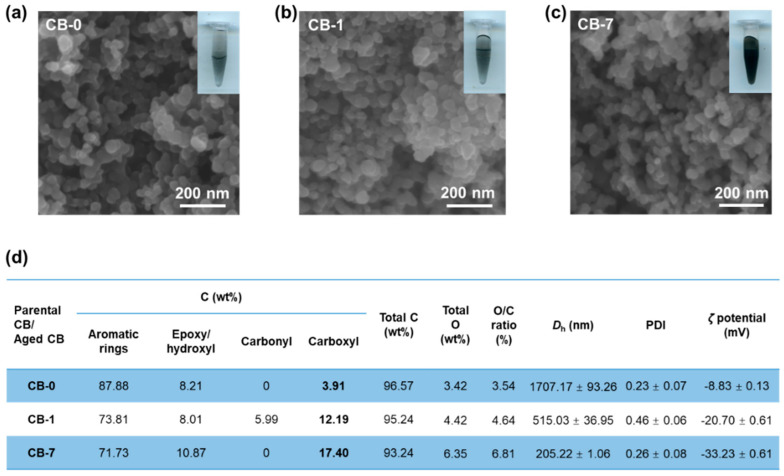
Characterization of parental and aged CB particles. (**a**–**c**) Representative SEM images of CB-0 (**a**), CB-1 (**b**), and CB-7 (**c**) particles, with inserts showing CB suspensions in DI water. (**d**) Representative physicochemical properties of parental CB and aged CB particles, *D*_h_ indicates the hydrodynamic diameter on the basis of particle size distribution analysis, and PDI represents the polydispersity index (n = 3). Abbreviations: CB, carbon black; *D*_h_, hydrodynamic diameter; PDI, polymer dispersity index.

**Figure 2 antioxidants-11-00754-f002:**
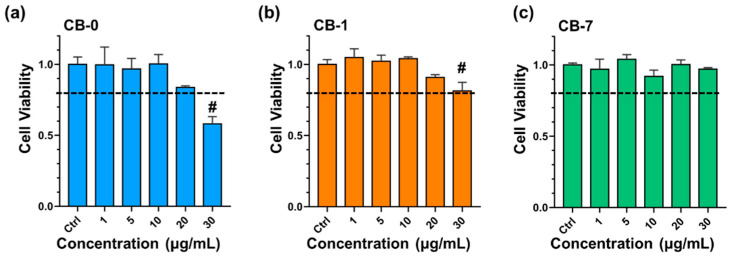
Cytotoxicity of parental and aged CB particles in J774A.1 cells. Cytotoxicity of CB-0 (**a**), CB-1 (**b**), and CB-7 (**c**) in J774A.1 cells after 24 h treatment at various concentrations (n = 6). All controls are untreated groups. Statistical significance between groups: (#) *p* < 0.001, relative to the untreated control. Abbreviations: CB, carbon black.

**Figure 3 antioxidants-11-00754-f003:**
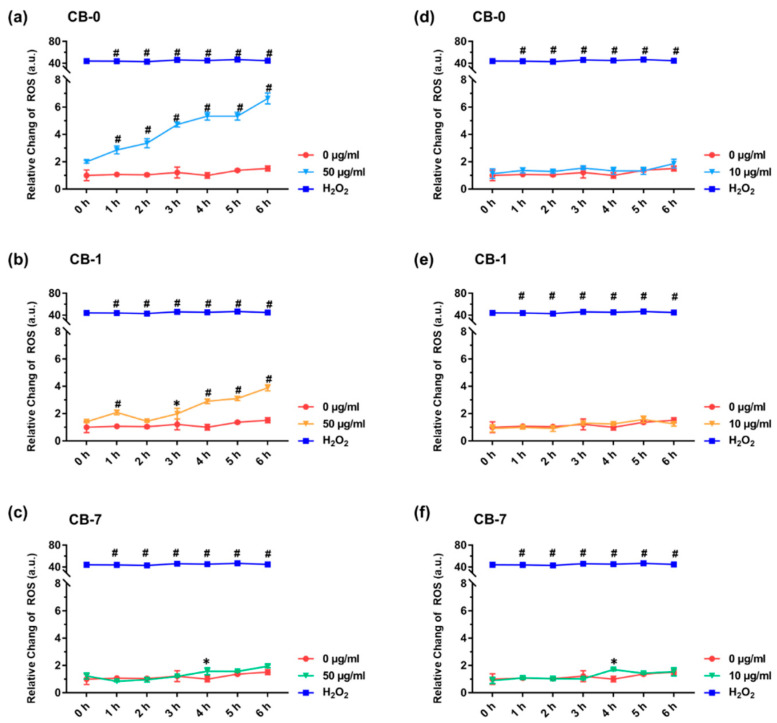
Determinations of intracellular ROS. Intracellular ROS production over time in J774A.1 cells upon CB particle treatment at 50 µg/mL (**a**–**c**) and 10 µg/mL (**d**–**f**) (n = 3). The untreated group was used as the negative control and 0.2% H_2_O_2_ was used as the positive control. All controls are untreated groups. Statistical significance between groups: (*) *p* < 0.05 and (#) *p* < 0.001, relative to the untreated control. Abbreviations: ROS, reactive oxygen species.

**Figure 4 antioxidants-11-00754-f004:**
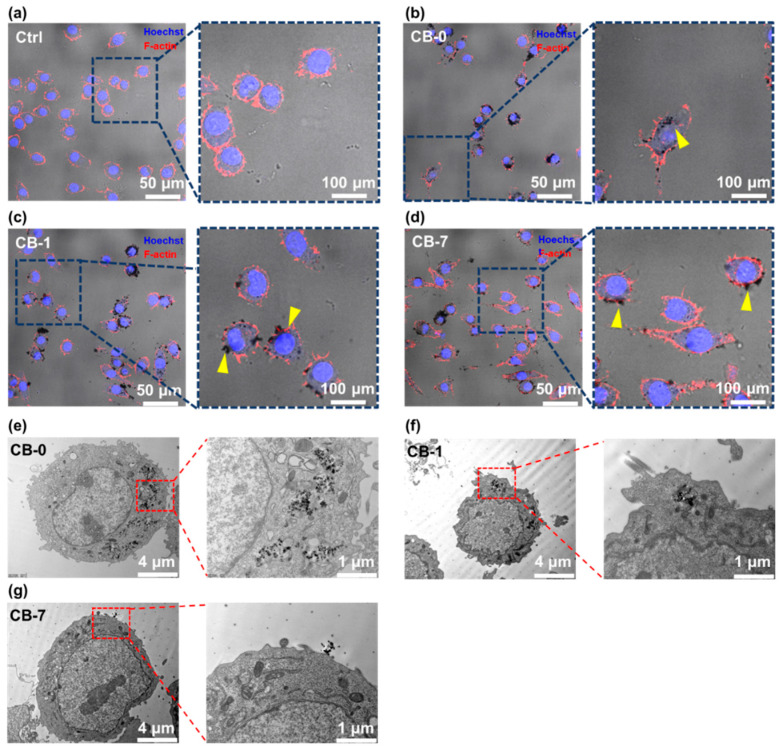
Interactions between CB particles and macrophages. (**a**–**d**) Confocal laser scanning microscopy images of J774A.1 macrophages after exposure to CB particles (10 µg/mL) at 24 h. The red and blue fluorescence signals represent the cytoskeleton (F-actin) and nucleus, respectively. The particle aggregates are denoted by yellow arrowheads. (**e**–**g**) TEM images displaying the intracellular localization of CB particles in J774A.1 cells upon treatment at 10 μg/mL for 12 h. Abbreviations: CB, carbon black.

**Figure 5 antioxidants-11-00754-f005:**
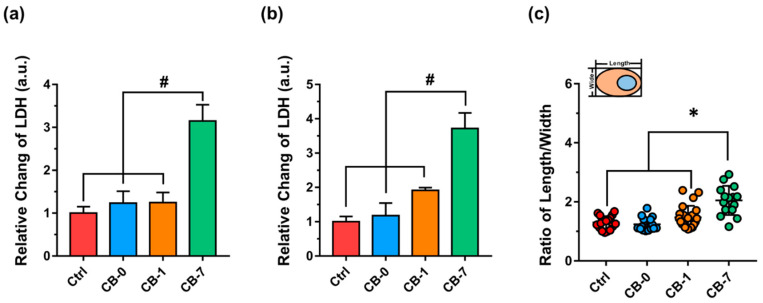
Cellular morphological changes by CB particles in macrophages. (**a**,**b**) The relative release of LDH from J774A.1 cells in response to CB particles at 5 µg/mL (**a**) and 10 µg/mL (**b**) for 24 h (n = 3). (**c**) Statistics of length to width ratio of J774A.1 macrophages after exposure to CB particles (10 µg/mL) at 24 h from confocal images, with a schematic diagram illustrating the statistical method of the ratio of length/width. All controls are untreated groups. Statistical significance between groups: (*) *p* < 0.05 and (#) *p* < 0.001, relative to the untreated control. Abbreviations: CB, carbon black; LDH, lactate dehydrogenase.

**Figure 6 antioxidants-11-00754-f006:**
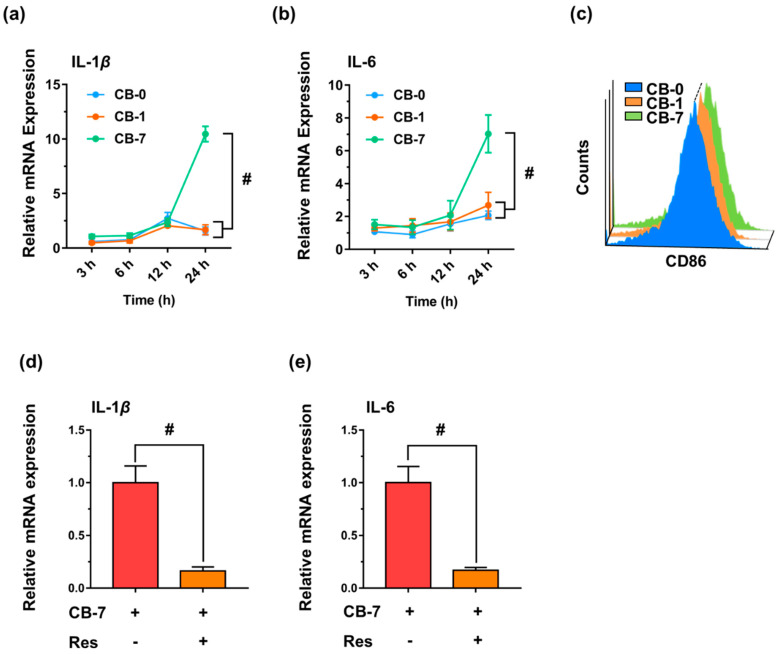
Inflammatory responses induced by CB particles in macrophages. (**a**,**b**) Gene expression levels of IL-1*β* and IL-6 in J774A.1 macrophages after exposure to CB particles at different time points (10 µg/mL, n = 6) relative to the untreated control. (**c**) CD86 expression level in J774A.1 macrophages after exposure to CB particles (10 µg/mL) for 24 h, as determined by flow cytometry analysis. (**d**,**e**) IL-1*β* and IL-6 expression levels in J774A.1 macrophages after exposure to CB particles for 24 h with or without pretreatment with the Res (at 1 μM, n = 6), as characterized by qRT-PCR analysis relative to the untreated CB-7 group. All controls are untreated groups. Statistical significance between groups: (#) *p* < 0.001. Abbreviations: CB, carbon black; IL, interleukin; Res, Resatorvid.

**Figure 7 antioxidants-11-00754-f007:**
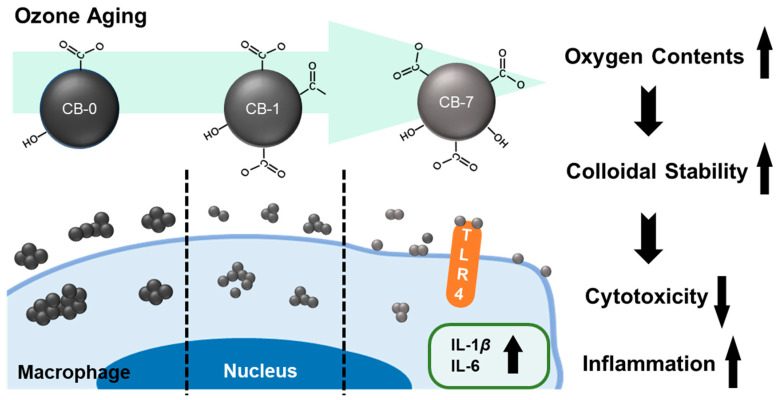
A schematic diagram illustrating the ageing process in altering the cytotoxicity profiles of CB particles in macrophages. Abbreviations: CB: carbon black; IL, interleukin; TLR4, toll-like receptor 4.

## Data Availability

All data are contained within the article and Appendix A.

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
