# Peer review of "Ageing Significantly Alters the Physicochemical Properties and Associated Cytotoxicity Profiles of Ultrafine Particulate Matters towards Macrophages"

_antioxidants, 2022, doi:10.3390/antiox11040754_

Round 1

Reviewer 1 Report

The paper by Yan et al investigates the cytotoxicity of particulate matter, subjected to an experimental ageing process, against macrophages, providing a new evidence of air pollution-associated health risks.

Despite its scientific soundness, the manuscript totally lacks an analysis of oxidative stress after CB exposure. The elucidation of oxidative mechanisms is a goal of the journal “Antioxidants”, so it is worth addressing. Therefore, a determination of oxidative stress markers (i.e., ROS generation, mitochondrial membrane potential, etc.) is necessary to consider this study consistent to be published in this journal.

Other minor points need to be addressed by Authors:

1) The title of paper is confounding. Please, make it clearer to let the reader catch the topic of this study immediately.

2) Despite the fine use of complex English language, Authors should simplify it, where possible, to make the reading easier and more fluent.

3) About “Materials and Methods” section:

- Please, specify the location of the company “Degussa Inc. Corp. (line 77) immediately after naming it and delete the full stop in each paragraph title.

- Please, report the cellular density seeded for each experiment.

- In paragraph 2.3, please specify that J774A.1 is a macrophage cell line and, in addition, indicate the CB tested concentrations in line 106 of paragraph 2.4.

- Paragraph 2.11 should follow the 2.9 one since both analyses were performed by qPCR.

4) About “Results and Discussion” section:

- The term “concentration” refers to in vitro models and “dose” to in vivo models. Therefore, correct this mistake both in line 218 and in Figure 2. In addition, CB cytotoxicity to J774A.1 cells is not concentration-dependent, since cell viability is not reduced in proportion to the increase of CB particle concentration, as observed in Figure 2.

- Please, specify the statistical comparison groups in each figure caption.

- Please, separate Figures 4a and 4b from 4c, d, e since the former do not concern the inflammatory process induced by CB particles on macrophages, as reported in the caption of figure.

Author Response

Reviewer Comments:

Reviewer #1:

The paper by Yan et al investigates the cytotoxicity of particulate matter, subjected to an experimental ageing process, against macrophages, providing a new evidence of air pollution-associated health risks.

Reply: We thank the reviewer for the positive comments.

Despite its scientific soundness, the manuscript totally lacks an analysis of oxidative stress after CB exposure. The elucidation of oxidative mechanisms is a goal of the journal “Antioxidants”, so it is worth addressing. Therefore, a determination of oxidative stress markers (i.e., ROS generation, mitochondrial membrane potential, etc.) is necessary to consider this study consistent to be published in this journal.

Reply: We appreciate the reviewer’s comments and we have added the ROS generation experiment to support our study. According to the results, exposure to CB particles would slightly induced ROS generation. However, there was no significant different among several materials. We have added the data in the SI and more discussions about the results in the manuscript (Line 268-271).

Figure S3 Determinations of intracellular ROS. Intracellular ROS production over time in J774A.1 cells upon CB particle treatment at 10 µg/mL (n = 3). Untreated cell group was used as the negative control and 0.2% H2O2 group was used as the positive control. Abbreviations: ROS, reactive oxygen species.

“The ROS generation was assayed and there is no significant difference among parental and aged CB particles (Figure S3).”

Other minor points need to be addressed by Authors:

1) The title of paper is confounding. Please, make it clearer to let the reader catch the topic of this study immediately.

Reply: We appreciate the reviewer’s comments and we have revised the title.

“Ageing significantly alters the physiochemical properties and the cytotoxicity profiles of ultrafine particulate matters towards macrophages”

2) Despite the fine use of complex English language, Authors should simplify it, where possible, to make the reading easier and more fluent.

Reply: We thank the reviewer for the thoughtful suggestions. We have revised and polished the whole manuscript through Language editing services from MDPI.

3) About “Materials and Methods” section:

- Please, specify the location of the company “Degussa Inc. Corp. (line 77) immediately after naming it and delete the full stop in each paragraph title.

Reply: We have added detailed information in Methods and revised accordingly.

“Degussa Inc. Corp. (China) provided commercial CB particles (Printex U). The ozonation of CB particle was accomplished using a well-established process, as reported in previous work [10,13].”

- Please, report the cellular density seeded for each experiment.

Reply: We have added detailed information in Methods.

Line135-136 “For cell viability assessment, after seeding in a 96-well (1.0 × 104 J774A.1 cells/well, Corning, USA) plate overnight, J774A.1 cells were treated with CB particles at various concentrations (i.e., 1, 5, 10, 20, 30 μg/ml) for 24 h.”

Line144-146 “After seeding in glass-bottom dishes (3.0 × 105 J774A.1 cells/well, Thermo Fisher Scientific, USA) overnight, J774A.1 cells were treated with CB particles at 10 µg/mL for 12 h.”

Line154-156 “After seeding in a 6-well plate (3.0 × 105 J774A.1 cells/well) overnight, J774A.1 cells were exposed to CB particles at 10 μg/mL for 12 h and then harvested with phosphate-buffered saline (PBS).”

Line169-170 “After seeding in a 6-well plate (3.0 × 105 J774A.1 cells/well) overnight, J774A.1 cells were exposed to CB particles at 10 μg/mL for 24 h.”

Line185-186 “After seeding in a 6-well (3.0 × 105 J774A.1 cells/well) plate overnight, J774A.1 cells were exposed to CB particles at 10 μg/mL for 24 h.”

Line199-200 “For pretreatment of the inhibitor against Toll-like receptor 4 (TLR4) signaling, J774A.1 cells were seeded in a 6-well plate (3.0 × 105 J774A.1 cells/well) overnight and then pretreated for 6 h with the inhibitor Resatorvid (Res, 1 μM, MCE, USA) to suppress TLR4 signaling, followed by CB particles exposure at 10 μg/mL for another 24 h.”

- In paragraph 2.3, please specify that J774A.1 is a macrophage cell line and, in addition, indicate the CB tested concentrations in line 106 of paragraph 2.4.

Reply: We have added the relevant information on J774A.1 cells and added detailed information in Methods (Line 118-119 and Line 137).

“The murine macrophage cell line J774A.1 and RAW264.7 were obtained from the Shanghai Cell Bank of Type Culture Collection (China).”

“For cell viability assessment, after seeding in a 96-well (1.0 × 104 J774A.1 cells/well, Corning, USA) plate overnight, J774A.1 cells were treated with CB particles at various concentrations (i.e., 1, 5, 10, 20, 30 μg/ml) for 24 h.”

- Paragraph 2.11 should follow the 2.9 one since both analyses were performed by qPCR.

Reply: We have revised the sections accordingly.

4) About “Results and Discussion” section:

- The term “concentration” refers to in vitro models and “dose” to in vivo models. Therefore, correct this mistake both in line 218 and in Figure 2. In addition, CB cytotoxicity to J774A.1 cells is not concentration-dependent, since cell viability is not reduced in proportion to the increase of CB particle concentration, as observed in Figure 2.

Reply: We have corrected the expression in the manuscript. To make the results more clearly, we have revised the description accordingly (Line 257-259).

“As shown in Figure 2, CB-0 and CB-1 particles showed concentration-dependent cytotoxicity to J774A.1 cells, while no significant cytotoxicity was found to J774A.1 cells upon CB-7 treatment.”

- Please, specify the statistical comparison groups in each figure caption.

Reply: We have added the information accordingly.

- Please, separate Figures 4a and 4b from 4c, d, e since the former do not concern the inflammatory process induced by CB particles on macrophages, as reported in the caption of figure.

Reply: We have revised the appearance of the figures accordingly.

Figure 4 Cellular change induced structure effect by CB particles in macrophages. (a-b) The relative release of LDH from J774A.1 cells in response to CB particles at 5 µg/mL (a) and 10 µg/mL (b) for 24 h (n = 3). (c) Statistics of length to width ratio of macrophages after exposure to CB particles (10 µg/mL) at 24 h from confocal images with a schematic diagram illustrating the statistical method of ratio of length/wide. All controls are untreated groups. Statistical significance between groups: (*) P < 0.05 and (#) P < 0.001,relative to untreated control. Abbreviations: CB, carbon black; LDH, lactate dehydrogenase.

Figure 5 Inflammatory reaction induced by aged CB particles in macrophages. (a-b) Gene expression levels of IL-1β and IL-6 in macrophages after exposure to CB particles at different time points (10 µg/mL, n = 6) relative to untreated control. (c) CD86 expression level in macrophages after exposure to CB particles (10 µg/mL) for 24 h, as determined by flow cytometry analysis. (d-e) IL-1β and IL-6 expression levels in macrophages after exposure to CB particles for 24 h with or without pretreatment with the Res (1 μM, n = 6), as characterized by qRT-PCR analysis relative to untreated CB-7 group. All controls are untreated groups. Statistical significance between groups: (#) P < 0.001. Abbreviations: CB, carbon black; IL, interleukin; Res, Resatorvid.

Reviewer 2 Report

  1. The authors have used nano-sized carbon black particles expose to ozone (“ageing”) for different periods to treat a cultured macrophage cell line, and then review changes in the structure and function of the macrophages with a variety of tools, methods and assays used. In the event they present some interesting data showing that at time zero the particles were phagocytosed but cells reduced in viability, while in “aged” macrophages there was markedly less phagocytosis and improved survival, but the macrophages showed a number of cell membrane abnormalities. These changes were mediated by oxidants on the surface of treated particles, and inhibitable by blocking the cell surface receptor TLR4. In this oxidative, TLR4-mediated processs there is a switch in the macrophage population towards a M1 functional phenotype. The work seems to have been done carefully and competently, and the data and figures are convincing.
  1. The main problem is that although the text can be followed, just about, the English in the paper is generally poor, quite difficult to read overall, and in places quite garbled. It needs a thorough re-write by someone well-versed in scientific writing in English.
  2. Throughout, there is a tendency for the authors to assume that the reader knows as much about the background and Methods as they do; this adds to the difficulties. Examples include: not stating upfront what “ageing” in this context was all about (initially I thought it was something to do with ageing of the cells); not immediately saying what the J774A.1 cells were; in Methods the heading for each sub-paragraph should include what signal was being sought…what is this assay/measure ABOUT?
  1. The Discussion is rather limited. For example, I would like to know: a)The ultra-fine particles used in this study are a lot smaller than the particles usually measured in epidemiology (PM5); can they discuss the relevance of their work in this wider context?     b) In respiratory disease related to oxidants, and I`m thinking especially of the airways in COPD, the phenotypic switch described in the most directly-exposed luminal macrophages has been said to be towards M2 (Eapen etal Sci Reports Nature 2017). I think the literature deserves more discussion in light of their finding, and in particular their potential clinical relevance is worth dealing with.      c) It would be worth constructing a dose-response curve for (some of) the effects shown, especially to see if there is a threshold effect relevant to potential public health advice.    d) What is their cell-viability assay actually measuring?
  2.  The citations seem to be heavily biased towards the Chinese literature, and I don`t think that this can properly represent the "state of art". 

Author Response

Comments and Suggestions for Authors

  1. The authors have used nano-sized carbon black particles expose to ozone (“ageing”) for different periods to treat a cultured macrophage cell line, and then review changes in the structure and function of the macrophages with a variety of tools, methods and assays used. In the event they present some interesting data showing that at time zero the particles were phagocytosed but cells reduced in viability, while in “aged” macrophages there was markedly less phagocytosis and improved survival, but the macrophages showed a number of cell membrane abnormalities. These changes were mediated by oxidants on the surface of treated particles, and inhibitable by blocking the cell surface receptor TLR4. In this oxidative, TLR4-mediated process there is a switch in the macrophage population towards a M1 functional phenotype. The work seems to have been done carefully and competently, and the data and figures are convincing.

Reply: We thank the reviewer for the positive comments. We have answered your specific questions/comments below.

  1. The main problem is that although the text can be followed, just about, the English in the paper is generally poor, quite difficult to read overall, and in places quite garbled. It needs a thorough re-write by someone well-versed in scientific writing in English.

Reply: We thank the reviewer for the thoughtful suggestions. We have revised and polished the whole manuscript through Language editing services from MDPI.

  1. Throughout, there is a tendency for the authors to assume that the reader knows as much about the background and Methods as they do; this adds to the difficulties. Examples include: not stating upfront what “ageing” in this context was all about (initially I thought it was something to do with ageing of the cells); not immediately saying what the J774A.1 cells were; in Methods the heading for each sub-paragraph should include what signal was being sought…what is this assay/measure ABOUT?

Reply: We appreciate the reviewer for the suggestion and we have added more descriptions to clarify the definitions on “ageing” and cell lines accordingly. We have also revised the Methods as the reviewer suggested (Line 44-47, and line 255-257).

“It should be noted that the ambient environment contains complicate features, such as SO2, O3, NOx and so on, and would react with PMs, and the long-term process of aerosol heterogeneous reaction of PMs in atmospheric environment would be often referred to ageing [7].”

“Therefore, to corroborate our hypothesis, macrophages were assayed upon treatment of different CB particles, and J774A.1 cells, as a typical model of macrophage [47], were thereafter chosen for this purpose.”

  1. The Discussion is rather limited. For example, I would like to know: a)The ultra-fine particles used in this study are a lot smaller than the particles usually measured in epidemiology (PM5); can they discuss the relevance of their work in this wider context?

Reply: We appreciate the reviewer for the thoughtful suggestion and we have added the discussions as follows (Line 335-337).

“Of various air pollutants, ambient fine particles, especially UFPs (i.e., PM0.1), are thought to be the most concerning in terms of their detrimental health effects to the lung, since they can escape from the mucous barrier and enter the lungs [18,53].”

  1. b) In respiratory disease related to oxidants, and I`m thinking especially of the airways in COPD, the phenotypic switch described in the most directly-exposed luminal macrophages has been said to be towards M2 (Eapen etal Sci Reports Nature 2017). I think the literature deserves more discussion in light of their finding, and in particular their potential clinical relevance is worth dealing with.

Reply: We have added the discussions as follows (Line 354-356).

“Note that alterations of the immune balance in the lung have been proposed as reasons of various diseases, such as COPD, acute lung injury and pulmonary hypertension [54-56].”

  1. c) It would be worth constructing a dose-response curve for (some of) the effects shown, especially to see if there is a threshold effect relevant to potential public health advice.

Reply: We appreciate the reviewer’s comment.

The dose-response curve for the inflammatory effects of CB-7 on J774A.1 cells was investigated. It should be noted that to mimic a more real exposure scenario, we chose the sub-lethal concentrations (i.e., 5 and 10 µg/mL) for experiments, which is still much higher than that in the realistic exposure condition (DOI: 10.1002/smll.202000603). Hence, the complex threshold effect of CB particles that is relevant to potential public health advice would be carefully studied in the future.

IL-1β and IL-6 expression levels in J774A.1 macrophages after exposure to CB particles for 24 h under various concentrations. Statistical significance between groups: (#) P < 0.001,relative to untreated control.

  1. d) What is their cell-viability assay actually measuring?

Reply: We evaluated the cell viability upon CB particle exposure by using the cell counting kit (CCK-8). Briefly, CCK-8 tetrazolium salt can be reduced by cellular dehydrogenases to an orange formazan product that is soluble in tissue culture medium. The amount of formazan produced is directly proportional to the number of living cells. Hence, the cell viability assay actually measured the amount of living cells, compared with the control group.

  1. The citations seem to be heavily biased towards the Chinese literature, and I don`t think that this can properly represent the "state of art".

Reply: We appreciate the reviewer’s suggestion and we have added more references as Ref. 5, 12, 16, 23 and 33.

“5. Matawle, J.L.; Pervez, S.; Shrivastava, A.; Tiwari, S.; Pant, P.; Deb, M.K.; Bisht, D.S.; Pervez, Y.F. PM2.5 pollution from household solid fuel burning practices in central India: 1. Impact on indoor air quality and associated health risks. Environ Geochem Health 2017, 39, 1045-1058, doi:10.1007/s10653-016-9871-8.

  1. Bates, J.T.; Fang, T.; Verma, V.; Zeng, L.; Weber, R.J.; Tolbert, P.E.; Abrams, J.Y.; Sarnat, S.E.; Klein, M.; Mulholland, J.A.; et al. Review of Acellular Assays of Ambient Particulate Matter Oxidative Potential: Methods and Relationships with Composition, Sources, and Health Effects. Environ Sci Technol 2019, 53, 4003-4019, doi:10.1021/acs.est.8b03430.
  2. Samoli, E.; Rodopoulou, S.; Schneider, A.; Morawska, L.; Stafoggia, M.; Renzi, M.; Breitner, S.; Lanki, T.; Pickford, R.; Schikowski, T.; et al. Meta-analysis on short-term exposure to ambient ultrafine particles and respiratory morbidity. Eur Respir Rev 2020, 29, doi:10.1183/16000617.0116-2020.
  3. Elsabahy, M.; Wooley, K.L. Cytokines as biomarkers of nanoparticle immunotoxicity. Chem Soc Rev 2013, 42, 5552-5576, doi:10.1039/c3cs60064e.
  4. Murray, P.J. Macrophage Polarization. Annu Rev Physiol 2017, 79, 541-566, doi:10.1146/annurev-physiol-022516-034339.”

Reviewer 3 Report

This paper studied the ageing process on UFP physiochemical changes and showed that CB particle ageing altered the cytotoxicity profiles and the inflammatory reactions on macrophages. They emphasized the species of surface O-containing groups dictated the bioreactivity and cyto-compatibility of aged CB particles towards macrophages.  This result is interesting but appears to be lack of sufficient data to support the above conclusion. Besides, the presentation of this paper needs to be significantly improved. There are many grammar errors and in-concise descriptions that need to be corrected/improved.  The list below are examples only, not an exhaustive list.  The authors need to undertake a thorough check/proofreading.  

Line 28: change “enter” to ‘entering”

Lines 36-37: “is so far limited understanding” is Grammarly incorrect.  Please rephrase.

Line 43: change “upon” to “among”.  There are multiple misuses of “upon” in this manuscript.  A thorough grammar check is needed. 

Line 46: “documented”, not “documenting”

Lines 49-50: “is dependent on” or “depends on”

Line 51:  There are many errors in present/past tense in the manuscript.  In this place, change “involves” to ‘involved”.

Line 70: change “upon” to “during the”

Line 71: another example of misuse of “upon” here. Change “upon” to “from”.

Line 138: “Determined of lactate dehydrogenase (LDH) release”?  Please clarify.

Lines 171-180: These sentences are methods not results.  They should either be removed or put into methods, and simply start with “Figures 1a-c shows the SEM images ….”.

Line 284: “we thereafter examined the TLR4 signaling pathway elaborate M1 polarization by CB-7.” I have difficulty in understanding what this sentence is to express. Please rephase.

Lines 289-292, Fig. 4g: inconsistent between the text and figure.  There is no difference in Fig. 4g among CB0, CB1, and CB7.

Line 324: change “interaction … to “ to “interaction … with “

Lines 317-325: “Nonetheless, the high content of carboxyl groups …. balance in lung (Figure 5).”  These three sentences are redundant and unnecessarily wordy. Please simplify it and make it more readable in a logical way.

Line 235: “unraveling”, not “unravel”.

Author Response

Comments and Suggestions for Authors

This paper studied the ageing process on UFP physiochemical changes and showed that CB particle ageing altered the cytotoxicity profiles and the inflammatory reactions on macrophages. They emphasized the species of surface O-containing groups dictated the bioreactivity and cyto-compatibility of aged CB particles towards macrophages. This result is interesting but appears to be lack of sufficient data to support the above conclusion. Besides, the presentation of this paper needs to be significantly improved. There are many grammar errors and in-concise descriptions that need to be corrected/improved. The list below are examples only, not an exhaustive list. The authors need to undertake a thorough check/proofreading. 

Reply: We appreciate the reviewer for the positive comments and we have revised the manuscript thoroughly. We have revised and polished the whole manuscript through Language editing services from MDPI.

Line 28: change “enter” to ‘entering”

Reply: We have revised it (Line 43).

“Once entering the atmosphere, PMs, especially ultrafine particulate matter (UFP), will float in the air in the long term due to their small sizes [4-6].”

Lines 36-37: “is so far limited understanding” is Grammarly incorrect.  Please rephrase.

Reply: We have revised it (Line 51-53).

“Although many studies suggested the physiochemical changes in PMs after the ambient ageing process, the ageing-determined toxicity changes of PMs is not clearly understood.”

Line 43: change “upon” to “among”.  There are multiple misuses of “upon” in this manuscript.  A thorough grammar check is needed.

Reply: We have revised it and checked the grammar (Line 59-61).

“Among the various functional cells in the reticuloendothelial systems, macrophages stand out as the first line of defense against foreign particles through playing an important role in particle transport and clearance [17].”

Line 46: “documented”, not “documenting”

Reply: We have revised it (Line 62-65).

“Documented by previous studies, the mechanisms of cytotoxicity to macrophages caused by UFPs were ascribed to reactive oxygen species (ROS) generation, the damage of the cell membrane and inflammatory reactions [22-27].”

Lines 49-50: “is dependent on” or “depends on”

Reply: We have revised it (Line 65-66).

“In addition, the oxidative stress and toxicity of UFPs towards cells largely dependent on the surface properties [28,29].”

Line 51:  There are many errors in present/past tense in the manuscript.  In this place, change “involves” to ‘involved”.

Reply: We have revised it and checked the tense (Line 66-69).

“For instance, Wei et al. showed that the O3 aging involved a change in the oxygen containing functional groups of CB particles, and the content of epoxy groups would induce higher ROS generation and then higher oxidative stress to erythroid cells [13].”

Line 70: change “upon” to “during the”

Reply: We have revised it (Line 85-87).

“Therefore, we aimed to investigate the physiochemical alterations of the model UFPs during the ambient ageing process with O3, and to interrogate how the ageing process changes the interaction and toxicity profiles of UFPs towards macrophages.”

Line 71: another example of misuse of “upon” here. Change “upon” to “from”.

Reply: We have revised it (Line 87-89).

“In addition, the influences on the immune states of lungs by UFPs with different surface properties were scrutinized.”

Line 138: “Determined of lactate dehydrogenase (LDH) release”?  Please clarify.

Reply: We have revised it to make more clearly (Line 183).

Determinations of lactate dehydrogenase (LDH) release for cell membrane integrity evaluation

Lines 171-180: These sentences are methods not results.  They should either be removed or put into methods, and simply start with “Figures 1a-c shows the SEM images ….”.

Reply: We have revised it. The description of aged CB was deleted (Line 223-227).

“Figures 1a-c shows the SEM image, the images revealed that the 2 aged CB particles appeared in uniform particles with a diameter around 30 nm, consistent with CB-0. The SEM images indicated that there was no significant change in both the morphologies and sizes after the ageing process.”

Line 284: “we thereafter examined the TLR4 signaling pathway elaborate M1 polarization by CB-7.” I have difficulty in understanding what this sentence is to express. Please rephase.

Reply: We have revised it (Line 326-327).

“According to the interaction of CB-7 to the cell membrane, we thereafter examined the roles of TLR4 signaling pathway that elaborate M1 polarization by CB-7.”

Lines 289-292, Fig. 4g: inconsistent between the text and figure.  There is no difference in Fig. 4c among CB0, CB1, and CB7.

Reply: The expression of CD86 in Fig. 4g among CB-0, CB-1 and CB-7 showed a slight difference, which indicated the M1 polarization of macrophages. We have revised the expression to make it more clearly (Line 321-323).

“In agreement with to the above data, the flow cytometry analysis also showed that the CD86 expression, namely M1 marker, in macrophages slightly increased in response to CB-7 treatment for 24 h (Figure 5c).”

Line 324: change “interaction … to “ to “interaction … with “

Reply: We have revised it (Line 356-360).

“Thus, these results unraveled a previously unrecognized role of surface properties in determining cytotoxicity, that is, the alterations in surface O-functionalities after the ageing process can alter the interaction patterns of particles with the macrophages, and then change the cytotoxicity and the immune balance in the lung (Figure 6).”

Lines 317-325: “Nonetheless, the high content of carboxyl groups …. balance in lung (Figure 5).”  These three sentences are redundant and unnecessarily wordy. Please simplify it and make it more readable in a logical way.

Reply: We have revised it (Line 351-355).

“Nonetheless, the high content of carboxyl groups would also affect the aggregation state, and then induced less endocytosis of CB-7 particles by macrophages. In fact, the CB-7 particles would attach onto the membrane and evoke the M1 polarization of macrophages through the TRL4 pathway. The CB-7 exposure would eventually induce severer inflammation outcomes and impaired the immune balance in the lung.”

Line 235: “unraveling”, not “unravel”.

Reply: We have revised it (Line 274-277).

“Given that the changes in surface physiochemical properties could influence the toxicity profiles of CB particles, we further conducted experiments to explain the potential detrimental influence on macrophages through unraveling the interactions between particles and cells.”

Reviewer 4 Report

With interest, I read the manuscript antioxidants-1626063.

Overall, an interesting paper based on a well-conducted study. There are, however, several major reservations that need to be addressed:

Remark-1. The whole study is based on in vitro investigation only.

Remark-2. Only cell line is used. Not primary cells.

Remark-3. In addition, only a single cell line is used, thus any possible cell line specific effects on the results cannot be excluded.

Remark-4. Only a murine but not human cell line is investigated.

Remark-5. The data content is (thus) limited.

To address those points, the Authors should (crucial recommendations):

Recommendation-1. Describe the limitations of this study.

Recommendation-2. Change the classification of this work from Article to Communication.

Recommendation-3. Describe further experimental steps for future studies.

Additional comments:

Comment-1. Lines 63-65. And asthma (PMID: 31904412).

Comment 2. All abbreviations used in the Figures or Tables and their legends should be explained in the legends.

Comment 3. Using appropriate gene names (https://www.ncbi.nlm.nih.gov/gene; in italics) while talkin on mRNA should be considered.

Author Response

With interest, I read the manuscript antioxidants-1626063.

Overall, an interesting paper based on a well-conducted study. There are, however, several major reservations that need to be addressed:

Remark-1. The whole study is based on in vitro investigation only.

Remark-2. Only cell line is used. Not primary cells.

Remark-3. In addition, only a single cell line is used, thus any possible cell line specific effects on the results cannot be excluded.

Remark-4. Only a murine but not human cell line is investigated.

Remark-5. The data content is (thus) limited.

Reply: We thank the reviewer for the positive comments. We have answered your specific questions/comments below.

To address those points, the Authors should (crucial recommendations):

Recommendation-1. Describe the limitations of this study.

Recommendation-2. Change the classification of this work from Article to Communication.

Recommendation-3. Describe further experimental steps for future studies.

Reply: We really appreciate the reviewer’s comments.

First, we have added more in vitro cell lines to support our findings. The cytotoxicity profiles of CB particles were evaluated in another type of murine macrophages, i.e., RAW 264.7 cells. In addition, the inflammation reactions induced by CB particles were observed on RAW 264.7 cells and murine bone marrow-derived macrophages (BMDMs). The data were provided in SI.

Figure S2 Cytotoxicity of parental and aged CB particles in RAW 264.7 cells. Cytotoxicity of CB-0 (a), CB-1 (b) and CB-7 (c) in RAW 264.7 cells after 24 h treatment at various concentrations (n = 6). All controls are untreated groups. Statistical significance between groups: (#) P < 0.001,relative to untreated control. Abbreviations: CB, carbon black.

Figure S5 Inflammatory reactions induced by CB particles in macrophages. IL-1β and IL-6 expression levels in RAW 264.7 cells (a-b) and BMDMs (c-d) after exposure to CB particles for 24 h, as characterized by qRT-PCR analysis (n = 6). All controls are untreated groups. Statistical significance between groups: (*) P < 0.05 and (#) P < 0.001,relative to untreated control. Abbreviations: CB, carbon black; IL, interleukin; BMDM, murine bone marrow-derived macrophage.

We have added the relevant discussions as follows (Line 266-268, and Line 314-323).

“In addition, the cytotoxicity profiles of CB particles on another type of murine macrophages, i.e., RAW 264.7 cells, were also evaluated (Figure S2) and the results consistently confirmed the greatest cytotoxicity of CB-0 on macrophages.”

“As shown in Figure 5a-b, the CB-7 particles enhance the greatest expression of IL-1β during 24 h exposure (approximately with 10.45-fold in relative to the untreated cells, P < 0.001). In addition, similar changes of IL-6 were demonstrated, supporting the findings on the severer inflammation reactions upon CB-7 treatment. For comparison, the expressions of IL-1β and IL-6 induced by CB particles in other model macrophages, i.e., RAW 264.7 cells and BMDMs, were also investigated, and the results corresponded to those of J774A.1 cells, as the CB-7 would induce the most severe inflammatory reactions (Figure S5). In agreement with to the above data, the flow cytometry analysis also showed that the CD86 expression, namely M1 marker, in macrophages slightly increased in response to CB-7 treatment for 24 h (Figure 5c).”

Second, we have also described the limitations of the study, which would be addressed in the future study, in the discussion parts. Based the supplementary data, we believed that the content of the manuscript would satisfy the request of the journal (Line 356-364).

“Thus, these results unraveled a previously unrecognized role of surface properties in determining cytotoxicity, that is, the alterations of surface O-functionalities after ageing process can alter the interaction patterns of particles with the macrophages, and then change the cytotoxicity and the immune balance in lung (Figure 6). It should be noted that our study focused on the in vitro experiments and the findings did warrant follow-up in vivo studies, for example. Nevertheless, our findings stressed the importance of the ambient ageing process for cytotoxicity and, more importantly, long-term inflammation reaction and the balance of lung immune micro-environment.”

Additional comments:

Comment-1. Lines 63-65. And asthma (PMID: 31904412).

Reply: We have added the reference as Ref. 36.

“36. Potaczek, D.P.; Miethe, S.; Schindler, V.; Alhamdan, F.; Garn, H. Role of airway epithelial cells in the development of different asthma phenotypes. Cell Signal 2020, 69, 109523, doi:10.1016/j.cellsig.2019.109523.”

Comment 2. All abbreviations used in the Figures or Tables and their legends should be explained in the legends.

Reply: We appreciate the comments and have added the explanations.

Comment 3. Using appropriate gene names (https://www.ncbi.nlm.nih.gov/gene; in italics) while talkin on mRNA should be considered.

Reply: We appreciate the comments and have added the detailed information in Table S1.

Table S1 The primers used in real-time quantitative reverse transcription-polymerase chain reaction (qRT-PCR) reactions in the current study.

Genes

Forward (5’-3’)

Reverse (5’-3’)

IL-1β a

GCAACTGTTCCTGAACTCAACT

ATCTTTTGGGGTCCGTCAACT

IL-6 b

CTGCAAGAGACTTCCATCCAG

AGTGGTATAGACAGGTCTGTTGG

CyC c

AAGAAGGCATGAACATTGTGGAAGC

CGGAAATGGTGATCTTCTTGCTGG

a IL-1β: interleukin- 1beta (XM_006498795.5)

b IL-6: interleukin-6 (NM_001314054.1)

c CyC: cyclophilin (NM_008907.2)

Round 2

Reviewer 1 Report

Despite the Authors have responded to most of my requests, the manuscript needs further revision. As already strongly recommended, since the elucidation of oxidative mechanisms is a goal of the journal “Antioxidants”, it is necessary that the manuscripts published in this journal address this issue.  Therefore, I invite the Authors to implement and discuss their findings correlating them with oxidative stress throughout the manuscript. Without this, despite its scientific soundness, this study cannot be considered consistent for publication in this journal.

For the same reasons as above, although results on ROS generation assay were not very encouraging, it would be more appropriate if they were moved to the main manuscript rather than to the supplementary materials. Moreover, Authors should discuss these findings extensively and corroborate them with relevant literature.

In lines 50-51 of the main manuscript, Authors documented previous studies reported that the mechanisms of cytotoxicity to macrophages caused by UFPs were ascribed to reactive oxygen species (ROS) generation, the damage of the cell membrane and inflammatory reactions. How Authors explain that in their study the carbon black (CB) particles do not significantly induce ROS generation?

Author Response

Comments and Suggestions for Authors

Despite the Authors have responded to most of my requests, the manuscript needs further revision. As already strongly recommended, since the elucidation of oxidative mechanisms is a goal of the journal “Antioxidants”, it is necessary that the manuscripts published in this journal address this issue.  Therefore, I invite the Authors to implement and discuss their findings correlating them with oxidative stress throughout the manuscript. Without this, despite its scientific soundness, this study cannot be considered consistent for publication in this journal (Line 355-358).

Reply: We thank the reviewer for the helpful comments. We have implemented additional experiments and added more discussion on oxidative stress and their related effects.

“Although the CB-7 particles exhibited the least cytotoxicity, CB-7 could induce the most serious inflammatory reactions in macrophages. Intriguingly, our results indicated no significant variation in ROS generation upon these different types of CBs under a sub-lethal concentration. Nonetheless, CB particles still induced other detrimental outcomes in macrophages, e.g., inflammation, especially CB-7. The remarkable biological effects of CB-7 might be due to the significant increase in carboxyl groups on the surface after ageing, increasing colloidal stability and cytocompatibility.”

For the same reasons as above, although results on ROS generation assay were not very encouraging, it would be more appropriate if they were moved to the main manuscript rather than to the supplementary materials. Moreover, Authors should discuss these findings extensively and corroborate them with relevant literature.

Reply: We thank the reviewer for the helpful comments. In spired by the comments from the reviewer, we performed additional experiments at different concentrations.

We have carried experiments to figure out the ROS generation under CB particle treatments, by increasing the concentration of CB particles. Briefly, under high concentration treatments (e.g., 50 µg/mL, Figure3a-c), the ROS generation would be significantly induced upon CB treatments, as CB-0 induced the greatest ROS production, followed by CB-1 and CB-7 (namely, CB-0 > CB-1 > CB-7). Reversely, the sub-lethal concentration in our study (10 µg/mL) induces little increase of ROS generation, as shown in Figure3d-f.

Figure 3. Determinations of intracellular ROS. Intracellular ROS production over time in J774A.1 cells upon CB particle treatment at 50 µg/mL (a-c) and 10 µg/mL (d-f) (n = 3). Untreated group was used as the negative control and 0.2% H2O2 was used as the positive control. All controls are untreated groups. Statistical significance between groups: (*) P < 0.05 and (#) P < 0.001, relative to untreated control. Abbreviations: ROS, reactive oxygen species.

As the reviewer’s suggestion, we have moved the ROS data as Figure 3 and added more discussions as follows (Line 270-280).

“For comparison to previous studies, the ROS generation was assayed. CB-0 induced the greatest ROS generation in J774A.1 cells at 50 µg/mL, followed by CB-1 and CB-7 (i.e., CB-0 > CB-1 > CB-7, Figure 3a-c). The results indicated that the CB-0 triggered the greatest oxidative stress in macrophages, consistent with the cytotoxicity profiles. Nevertheless, there was no significant variation in ROS generation upon these different types of CBs, compared with untreated control under a sub-lethal concentration (i.e., 10 µg/mL, Figure 3d-f). These findings suggested that a number of complex factors may decide the overall toxicity, as discussed in previous work [13,30]. Intriguingly, despite little ROS generation at the low exposure concentration, CB particles would still initiate other significant phenotypes in macrophages, such as inflammation, as described above.”

In lines 50-51 of the main manuscript, Authors documented previous studies reported that the mechanisms of cytotoxicity to macrophages caused by UFPs were ascribed to reactive oxygen species (ROS) generation, the damage of the cell membrane and inflammatory reactions. How Authors explain that in their study the carbon black (CB) particles do not significantly induce ROS generation?

Reply: As we replied above, the ROS generations would be significantly induced upon CB treatments under a high concentration rather than the sub-lethal concentration. The discrepancy may be due to differential responses in different cell lines, such as J774A.1 cells in the current study and divergent cells in other previous studies. For example, Kroll et al. found significant ROS generation (with about 2-fold increase relative to untreated control) under carbon black (CB) particle treatments in RAW264.7 cells [1]. Cao et al. used THP-1 cells and found about 3-fold ROS production after CB exposure relative to the untreated cells [2].

Second, the concentrations of CB particles also affected the results. For instance, Wei et al. found significant ROS generation under 50 µg/mL exposure in RAW264.7 cells [3] and Figarol et al. found significant ROS increase under 120 µg/mL exposure in RAW264.7 cells [4]. In fact, the sub-lethal concentration in our study (10 µg/mL) was much lower than that in other studies, but it is more comparable to realistic exposure scenario. Hence, the slight ROS inductions in our study is reasonable.

In addition, the sizes, morphologies and surface properties of CB particles will also affect the ROS generations in cells. Therefore, a number of complex factors may decide the oxidative stress induction, which warrants further detailed investigation.

Collectively, the CB particles in our study would not induce significant ROS generation in J774A.1 cells under a realistic concentration. We also clarified the results in main text (Line 270-280).

“For comparison to previous studies, the ROS generation was assayed. CB-0 induced the greatest ROS generation in J774A.1 cells at 50 µg/mL, followed by CB-1 and CB-7 (i.e., CB-0 > CB-1 > CB-7, Figure 3a-c). The results indicated that the CB-0 triggered the greatest oxidative stress in macrophages, consistent with the cytotoxicity profiles. Nevertheless, there was no significant variation in ROS generation upon these different types of CBs, compared with untreated control under a sub-lethal concentration (i.e., 10 µg/mL, Figure 3d-f). These findings suggested that a number of complex factors may decide the overall toxicity, as discussed in previous work [13,30]. Intriguingly, despite little ROS generation at the low exposure concentration, CB particles would still initiate other significant phenotypes in macrophages, such as inflammation, as described above.”

References

  1. Kroll, A.; Dierker, C.; Rommel, C.; Hahn, D.; Wohlleben, W.; Schulze-Isfort, C.; Gobbert, C.; Voetz, M.; Hardinghaus, F.; Schnekenburger, J. Cytotoxicity screening of 23 engineered nanomaterials using a test matrix of ten cell lines and three different assays. Part Fibre Toxicol 2011, 8, 9, doi:10.1186/1743-8977-8-9.
  2. Cao, Y.; Roursgaard, M.; Danielsen, P.H.; Moller, P.; Loft, S. Carbon black nanoparticles promote endothelial activation and lipid accumulation in macrophages independently of intracellular ROS production. PLoS One 2014, 9, e106711, doi:10.1371/journal.pone.0106711.
  3. Wei, S.; Qi, Y.; Ma, L.; Liu, Y.; Li, G.; Sang, N.; Liu, S.; Liu, Y. Ageing remarkably alters the toxicity of carbon black particles towards susceptible cells: determined by differential changes of surface oxygen groups. Environ Sci: Nano 2020, 7, 1633-1641, doi:10.1039/d0en00281j.
  4. Figarol, A.; Pourchez, J.; Boudard, D.; Forest, V.; Akono, C.; Tulliani, J.M.; Lecompte, J.P.; Cottier, M.; Bernache-Assollant, D.; Grosseau, P. In vitro toxicity of carbon nanotubes, nano-graphite and carbon black, similar impacts of acid functionalization. Toxicol In Vitro 2015, 30, 476-485, doi:10.1016/j.tiv.2015.09.014.

Reviewer 3 Report

The authors have addressed my concerns well.  

Author Response

The authors have addressed my concerns well.

Reply: We thank the reviewer for the positive comments.

Reviewer 4 Report

Thank you for addressing my comments well. I have no further reservations.

Author Response

Thank you for addressing my comments well. I have no further reservations.

Reply: We thank the reviewer for the positive comments.